# Histopathological Growth Pattern in Colorectal Liver Metastasis and The Tumor Immune Microenvironment

**DOI:** 10.3390/cancers15010181

**Published:** 2022-12-28

**Authors:** Claudia Zaharia, Torhild Veen, Dordi Lea, Arezo Kanani, Marina Alexeeva, Kjetil Søreide

**Affiliations:** 1Department of Pathology, Stavanger University Hospital, N-4068 Stavanger, Norway; 2Gastrointestinal Translational Research Group, Laboratory for Molecular Medicine, Stavanger University Hospital, N-4068 Stavanger, Norway; 3Department of Gastrointestinal Surgery, Stavanger University Hospital, N-4068 Stavanger, Norway; 4Department of Clinical Medicine, University of Bergen, N-7804 Bergen, Norway

**Keywords:** liver metastasis, colorectal cancer, hepatic metastases, tumor microenvironment, histopathology, growth pattern, immune system, invasive front, angiogenesis, recurrence, pathology, chemotherapy

## Abstract

**Simple Summary:**

Liver metastases occur in almost half of patients with colorectal cancer, either at the time of presentation (synchronous) or eventually as a metachronous recurrence of the disease. Approximately one quarter of patients with metastasis to the liver receive surgical resection with curative intent but recurrence is frequent and related to tumor biology. Understanding cancer biology is crucial to improve outcomes. In an attempt at unveiling novel targets for treatment, this paper investigates and summarizes current knowledge of the distinct growth patterns of tumor cells at the interface with normal liver tissue, defined as the histopathological growth pattern, and their interaction with immune cells in the microenvironment.

**Abstract:**

Almost half of all patients with colorectal cancer present with or eventually develop metastasis, most frequently in the liver. Understanding the histopathological growth patterns and tumor immune microenvironment of colorectal liver metastases may help determine treatment strategies and assess prognosis. A literature search was conducted to gather information on cancer biology, histopathological growth patterns, and the tumor immune microenvironment in colorectal liver metastases, including their mechanisms and their impact on clinical outcomes. A first consensus on histopathological growth patterns emerged in 2017, identifying five growth patterns. Later studies found benefits from a two-tier system, desmoplastic and non-desmoplastic, incorporated into the updated 2022 consensus. Furthermore, the tumor immune microenvironment shows additional characteristic features with relevance to cancer biology. This includes density of T-cells (CD8+), expression of claudin-2, presence of vessel co-option versus angiogenesis, as well as several other factors. The relation between histopathological growth patterns and the tumor immune microenvironment delineates distinct subtypes of cancer biology. The distinct subtypes are found to correlate with risk of metastasis or relapse, and hence to clinical outcome and long-term survival in each patient. In order to optimize personalized and precision therapy for patients with colorectal liver metastases, further investigation into the mechanisms of cancer biology and their translational aspects to novel treatment targets is warranted.

## 1. Introduction

Almost half of all patients with colorectal cancer present with or eventually develop metastatic disease, with the liver being the most frequent site of metastasis [1]. As systemic options advance and refinements in surgical techniques evolve, an increasing number of patients are offered liver resection or adjunct interventional therapy [2,3]. Several treatment options have emerged. These treatments range from more effective chemotherapy and biological therapy targeting specific pathways or angiogenesis, to more recently the application of immune checkpoint inhibitors [4]. In addition, several locoregional therapies, including ablation techniques, selective intern radiotherapy, and hepatic artery infusion have expanded treatment options and resection criteria [5,6,7,8,9]. 

Long-term survival at 5 years can be achieved in up to 50% of patients undergoing surgical resection of liver metastasis. Importantly, long-term survival is linked to actual cure in selected patients operated for liver metastasis [10,11,12,13]. In spite of this, there is considerable variation in patient selection for liver surgery [12]. 

Several attempts have been made at establishing prognostic models to predict outcome or likelihood of benefit from liver surgery, usually taking into account features such as size, number, and location of metastasis, as well as the time sequence from primary tumor treatment to development of liver metastasis [14]. Recently, molecular features such as KRAS mutations have been incorporated into risk score models [15,16,17]. Prognostic models used in hepatocellular carcinoma have also been tested for colorectal liver metastasis (CRLM), but are not in clinical use for this purpose [18,19]. It is not entirely clear how each of the factors in any given model contributes to prognostication, as there are time-dependent and biological variations in outcomes [20,21,22]. However, it is indisputable that there are “good, bad and ugly” CRLM, as manifested in clinical phenotypes and biological behavior (Figure 1). At the core of this is the inherent biological variation and interplay between factors of the tumor immune microenvironment [23,24]. 

In this study, we investigate the current knowledge of the histopathological growth patterns at the liver–tumor interface and the tumor immune microenvironment in resected colorectal liver metastases.

## 2. A Brief History of Histopathological Evaluation of Colorectal Liver Metastasis

The structured evaluation by microscopy of resected liver specimens has evolved over time, mostly with a focus on confirming the diagnosis and assessing resection margins. However, over the past decade, a keen and evolving interest in specific tumor growth patterns of liver metastases and their prognostic role has emerged. Indeed, a key 2013 paper coined the question, “Histopathological evaluation of liver metastasis: what should be done?” [25]. The authors found that histological features such as intrahepatic spread and resection margins, as well as tumor response to neoadjuvant chemotherapy, could represent promising prognostic factors that might influence subsequent decision making. A definite negative correlation between survival and invasion into the portal vein and lymphatics was found after reviewing several studies and meta-analyses. In their review, the authors also found resection margin to be an important prognostic factor, but with considerable controversy regarding the width of the presumed negative margins. Lastly, they found several studies evaluating tumor response to pre-operative chemotherapy, but different grading systems were used, and definite recommendations could not be made. The presence of a fibrous pseudo-capsule associated with improved survival is briefly mentioned, although potential growth patterns are not discussed further [25]. Of interest, the report elicited a response from the “Liver Metastasis Research Network”, which referred to the early description of three distinct histopathological growth patterns (HGP) at the interface between cancer cells and normal liver tissue [26,27,28]. Despite early suggestions of a correlation between HGP and prognosis, reporting growth patterns has not been universally implemented in clinical practice, most likely due to a lack of clinical consensus and standardized methodology [29]. 

## 3. Consensus Work on Histopathological Growth Patterns

The prognostic significance of HGPs has been established in several studies, showing superior overall survival in patients with desmoplastic growth pattern [30,31,32,33,34,35,36]. However, differences in methodology, as well as in sample size and reported incidence of the different patterns, emphasized the need for a standardized method of assessment.

The first consensus work on HGP of liver metastases emerged in 2017 [37]. The consensus outlined a decision algorithm and standard method to assess growth patterns of liver metastases. Three common growth patterns were identified: desmoplastic, replacement, and pushing, as well as two rare growth patterns: sinusoidal and portal. The desmoplastic growth pattern shows a fibrous rim that separates tumor cells from hepatocytes. In the replacement growth pattern, cancer cells occur in continuity with hepatocyte plates, whereas the pushing pattern is characterized by tumor tissue pushing against the liver tissue without a desmoplastic reaction. In the rare setting of a sinusoidal pattern, cancer cells grow in the lumina of sinusoidal vessels, adjacent to the hepatocyte plates. Portal growth pattern shows tumor cells in the portal tracts and septa, and/or within the lumina of biliary duct branches. HGP should be assessed at the tumor–liver interface by light microscopy of good quality H&E sections of resection specimens from CRLM. For metastases with a heterogeneous growth, a relative percentage of each growth pattern that constitutes more than 5% of the total length of the interface should be reported. In the case of multiple CRLM, a HGP is given for each metastasis [37]. 

In this first consensus work, 12 independent observers scored a set of 159 liver metastases. With regards to interobserver variability, the investigators found good-to-excellent correlations with the experts’ gold standard, with intraclass correlation coefficient values >0.5 for replacement and desmoplastic HGP. In a second set, an independent cohort of 374 patients with colorectal liver metastases, the impact of HGPs on overall survival after liver resection was assessed. A cutoff of >50% of the length of the tumor–liver interface was set, placing the patients in categories according to dominant HGP. A significant difference in survival was found in favor of desmoplastic HGP. Dividing patients into four categories based on the predominant growth pattern: desmoplastic, replacement, pushing, and mixed (Figure 2), was proposed for investigating the prognostic value of HGPs. Several validation studies followed. In one Dutch study, reliability and replicability were evaluated using a large dataset, with assessment of HGP concordance between tissue blocks of the same metastasis and tissue blocks from multiple metastases selected from patients with >2 synchronous metastases. Both the within (95%) and the between (90%) metastasis HGP concordance was high, with similar or better results in a validation set [35]. Diagnostic accuracy improved when scoring two blocks versus one block, but not when scoring three versus two blocks. After two training sessions, the interobserver agreement for both the pathologist (expert) and a Ph.D. candidate (novice) were excellent (kappa = 0.953 and kappa = 0.951, respectively) [35]. Notably, this study used a two-tiered histopathological assessment, dividing HGP into desmoplastic (100% desmoplastic growth) and non-desmoplastic (<100% desmoplastic growth). The investigators concluded that HGPs in CRLM exhibited little heterogeneity and could be determined with a high diagnostic accuracy, thus, making HGP a reliable and replicable histological biomarker in CRLM. In a further large retrospective multicenter cohort from the Netherlands and USA including over 780 patients, the correlation between CRLM with desmoplastic growth and superior overall survival was confirmed, even reporting an independent role of HGP compared with KRAS and BRAF mutation statuses [32].

A second, updated consensus guideline emerged in 2022 with the main goal of implementing histological growth patterns into the clinical decision process for treatment of patients with liver metastases [38]. The updated consensus established a new cutoff for HGP, based on the growing body of literature demonstrating the superior correlation of desmoplastic versus non-desmoplastic growth pattern with patient outcome over the previously proposed four categories (Figure 2) [32]. The consensus work featured a large multicentric cohort of 1931 patients that showed a mean overall survival of 88 months in patients with exclusively desmoplastic growth, compared with 53 months for the non-desmoplastic growth. At the same time, there were no differences in survival of patients with any of the other growth patterns. Similar findings in other tumor types suggest that the underlying biological mechanisms that shape the histological features of liver metastases are probably more liver specific and cancer-type independent [39,40,41]. The authors recommend that applying the two-tier algorithm in prognostic studies should not underscore the necessity of understanding tumor heterogeneity on a molecular/biological level. 

## 4. HGP Related to Tumor-Immune Microenvironment 

HGP and the anti-tumor immune response are related and may reflect the immune system’s ability to fence off cancer growth in the liver [32,33,42]. Three distinct immune phenotypes have been proposed in cancer: the immune-desert, the immune-inflamed, and the immune-excluded phenotype [43]. These phenotypes are characterized by the distribution and activity of T-cells in the tumor microenvironment and help to inform therapeutic strategies for initiating or restoring the anti-tumor immune response. The immune-desert phenotype shows scant infiltration with T cells in both tumor and stroma. This suggests the absence of a pre-existing antitumor immunity and that the generation of T cells is a rate-limiting step. In the immune-inflamed phenotype, T cells are present in the tumor bed, in near proximity of tumor cells, suggesting a pre-existing antitumor immune response that was arrested. Immune-excluded tumors show T-cells mainly in the tumor stroma, at the periphery of the tumor parenchyma. This suggests a pre-existing antitumor immune response that was rendered ineffective by lack of immune migration from the stroma to the tumor parenchyma. 

Desmoplastic HGP is associated with an inflammatory phenotype, which may explain the superior treatment response compared with the other growth patterns [44]. In one study, a higher rate of cytotoxic CD8+ T-cells was found in the desmoplastic type HGP, both in the peritumoral and intra-tumoral component [42]. The increased number of intraepithelial CD8+ T-cells, as opposed to the low stromal CD4+ T-cells, was suggested to represent a link to anti-tumor immunity and has been postulated to inhibit the development of micro-metastasis and further tumor spread. Several studies have reported high immune score in the desmoplastic type CRLM [33,45]. Corroborated risk models, incorporating clinical risk scores, immune score, and desmoplastic HGP, have shown increased overall survival at 5 years for patients in low-risk groups [12,32,33,45,46,47].

Replacement HGP is, on the other hand, associated with a non-immune phenotype and has a shorter recurrence-free survival [44]. The presence of replacement or mixed HGP in CRLM, combined with several clinicopathological features in a pathological score, proved to be, together with the immune score, most significant for predicting relapse after metastasis surgery [45]. High MHC-I expression and low CD3+ T-cells counts in replacement dominant HGP have been linked to risk of early recurrence [48]. Another study found that activated B-cells in the tumor front of CRLM imply a better prognosis, confirming the role of a favorable immune response in the tumor immune microenvironment. However, the presence of activated B cells in the tumor front was not correlated to histopathological features [49]. Among others, glycolytic cancer metabolism and the production of lactic acid resulting in an acidic environment seem to influence the immune system, such as the expression of PDL-1 in CD8+ T-cells [50,51]. A better understanding of the underlying mechanisms of the immune activating systems may facilitate enhanced immune-modulating treatment [4,23].

Claudin-2 was found to be predominant in liver metastases of patients with replacement HGP and was associated with poor overall survival in primary CRC and metastasis-free survival, as well as with the consensus molecular subtypes CMS 1 and CMS 3 [52]. In contrast, claudin-8 was found predominantly in desmoplastic CRLM. High claudin-2 levels have been detected in the plasma of patients with replacement CRLM. Therefore, claudin-2 has been proposed as a possible soluble biomarker that may predict development of replacement growth pattern in patients with CRLM [52]. If validated as biomarkers, claudin subtypes may hence serve as a relevant tool in designing optimal treatment strategies for patients with CRLM. 

Desmoplastic, and the more rare pushing HGP, induce angiogenesis and endothelial proliferation at the tumor–liver interface [53]. Many cancers induce angiogenesis to sustain growth (Figure 3), but this is not a universal phenomenon [54]. Of interest, the type of blood supply in CRLM can be deduced from the phenotypic HGP, to be either angiogenic or non-angiogenic (or vessel co-opting) [55]. 

Cancer cells in the replacement type HGP obtain their blood supply by vessel co-option [56]. Vessel co-option is enabled by changes in paracrine signals between cancer cells, hepatocytes, and the perisinusoidal environment [57]. Mouse models demonstrated that vessel co-option in liver metastatic lesions was attenuated with the knockout of Ang1 in comparison with the wild type mice [58]. It was shown that Ang1 stimulates the formation of vessel co-option via the Tie2-PI3K/-AKT-ARP2/3 pathway [59]. An important regulator of vessel co-option in replacement HGP type of CRLM is a runt-related transcription factor-1 (RUNX1), which is upregulated in the cancer cells and associated with hepatocyte displacement and replacement by cancer cells [59,60]. It is proposed that vessel co-option is correlated with the increased levels of infiltrating neutrophils via the RUNX1-Ang1 pathway [55]. However, the role of neutrophils in development and maintaining of vessel co-option for replacement type HGP remains unclear and requires further investigation. Vessel co-option is suspected to be a limiting factor in the effectiveness of anti-angiogenic treatment (e.g., bevacizumab; anti-VEGF inhibitor), and is assumed to contribute in acquiring resistance to both chemotherapy and immunotherapy [61,62]. 

## 5. Need for Better Understanding of the Cancer Biology of HGP in CRLM

Although several working hypotheses of tumor biology and the microenvironment of the different histopathological growth patterns have been proposed [38,44,47,61], there is currently no satisfactory biological explanation for the development of different histopathological growth patterns in CRLM. It may seem that cancer cells with replacement growth patterns adapt to the liver environment, behaving almost like hepatocytes, co-opting sinusoidal blood vessels and eliciting almost no immune response, whereas cancer cells with a desmoplastic growth pattern create their own environment in the liver-niche via processes of angiogenesis, inflammation, and fibrosis (Figure 3). 

Several factors seem to influence the histopathological growth of metastases, including cancer cell implantation site, presence or absence of coagulation and inflammation, response to liver injury with either fibrosis or regeneration, transcriptional reprogramming in replacement type metastases that express liver-specific genes, cancer cell motility, and angiotropic migration. Moreover, spatiotemporal and single-cell analyses demonstrate metabolic alterations and impact on immune-cell activation related to chemotherapy in the liver metastasis microenvironment [63]. 

Based on immunohistochemical analysis mapping spatial phenotypes of epithelial and stromal cells, two new theories have emerged [38]. First, that replacement growth might be the default growth pattern of CRLM that spontaneously or gradually transitions to desmoplastic growth. Co-existence of both desmoplastic and replacement types of growth in about 60% of CRLM suggests the possibility of transition from one growth pattern to another, rather than underlying changes in mutational constellation. Second, that activated SMA (smooth muscle actin)-positive fibroblasts expressing nerve growth factor receptor (NGFR) in the desmoplastic rim of CRLM, with a similar phenotype to portal fibroblasts, suggest a connection to resident portal tracts. Consequently, the HGP may, in part, be determined by cancer cell intrinsic properties such as cancer cell motility and differentiation. However, there is supporting evidence that growth patterns in CRLM should be perceived also as a response of the tumor microenvironment to stimuli and diverse epigenetic factors. 

One study of 29 paired primary and metastatic CRC cases investigated whether molecular changes in the primary tumor could predict HGP in liver metastasis, comparing primary CRC with desmoplastic HGP and replacement HGP liver metastases [64]. The investigators found that primary tumors with low tumor budding score and Crohn’s disease-like lymphoid infiltrates were associated with desmoplastic HGP and better overall survival. They also found different mutations in the primary tumors with the two different growth patterns, suggesting that mutations could be used to predict HGP growth pattern and prognosis. However, the number of cases was limited and the findings need validation in further series. On the other hand, in a multicentric study, HGPs had prognostic value in CRLM independent of KRAS or BRAF mutational status [32]. This supports the value of reporting HGPs in the pathology report alongside mutational status. 

No significant correlation has been found between tumor budding scores in primary tumors and their corresponding metastases in the liver [65]. Both tumor budding and poorly differentiated clusters (PDC) in CRLM are prognostic factors, with PDC indicated as an independent predictor of poor overall survival and disease-free survival. These factors are related to extrahepatic recurrence, suggesting that the formation of PDC may be a mechanism that allows progression to metastatic abilities [66]. 

The desmoplastic HGP seems to be associated with increased anti-tumor immune response in both the originating primary CRC and corresponding CRLM. A recent study looking at the relationship between primary cancer and HGPs of their corresponding metastases observed high-tumor-infiltrating lymphocytes (TIL) densities and Crohn’s disease-like lymphoid reaction in desmoplastic CRLM [67]. 

The presence of KRAS and SMAD4 mutations in CRLM suggest an initiating phenomenon in oligo-metastatic disease, defined as an indolent and long-term survival course of disease when it comes to comparing the molecular constellation of primary and metastatic CRC. Accumulation of mutational changes, mostly mutations in PI3KCA and BRAF, is observed in poly-metastatic disease. In addition, high counts of GrzB+ T cell-subsets have been documented in the tumor core of the metastases with an oligo-metastatic disease [68].

The Consensus Molecular Subtype (CMS) classification describes four CRC subtypes with distinct biological characteristics that show prognostic and potential predictive value in clinical setting [69]. Several attempts were made to evaluate the gene expression-based CMS of CRC in liver metastases, regardless of HGP type [70,71,72,73]. In spite of the intra-patient variations and effect of microenvironment, almost 90% of the studied CRLMs in one study expressed gene profiles that correspond to CMS2 and CMS4. Although patients with metastatic CMS1 or CMS3 have the shortest median overall survival, there was a relative depletion of CRLM with these subtypes in the study. Therefore, the authors acknowledge the results may be biased by patient selection [70].

## 6. Clinical Implications and Controversies of HGP in CRLM

Preoperative chemotherapy usually induces various degrees of treatment response assessed histopathologically by attributing a tumor regression grade (TRG). Notoriously, TRG has shown poor interobserver agreement independent of which grading system has been used [74]. The updated consensus guidelines define an ”escape” phenotype that should be reported in metastases resected after chemotherapy. This “escape” phenotype encompasses metastases that present signs of regression in the form of necrosis or fibrosis, and that exhibit a desmoplastic rim and small areas of replacement growth at the tumor periphery. Future studies are required to establish whether this feature will surpass TRG in assessing treatment response, or whether it will provide better understanding of underlying molecular mechanisms in presumed treatment induced HGP transition. 

One study proposed that chemotherapy induces desmoplastic growth in patients with replacement type CRLM, based on higher incidences of desmoplastic HGP in patients who received preoperative chemotherapy compared with chemo-naive patients in a randomized phase III clinical trial [75]. However, we do not know if this observation is caused by higher resistance to chemotherapy in desmoplastic metastases, or whether systemic treatment induces a desmoplastic rim in originally non-desmoplastic metastases [75]. These questions might be answered when reliable non-invasive modalities to predict HGP at different stages in the time sequence of treatment become available. 

Several studies suggest that systemic therapies may interfere and change histological growth patterns of CRLM [29,34,36,75,76]. Recurrent metastases after systemic treatment with bevacizumab (anti-VEGF inhibitor) often show a replacement growth pattern. This might support findings of previous studies that liver metastases of other tumor types (hepatocellular carcinoma, melanoma, and renal cell carcinoma) could have the ability to develop resistance to anti-VEGF drugs by switching from an angiogenic to a vessel co-opting growth pattern [62].

In an Italian cohort, the replacement type was associated with local recurrence and a higher risk for margin invasion, suggesting that a wider resection margin (at least >1 mm) should be attempted in this setting [30]. The authors assessed HGP in a three-tiered fashion, where the dominant (>50%) growth pattern—desmoplastic, pushing, or replacement—was recorded. In cases with mixed growth, the growth pattern with expected worse prognosis was assigned. In their study of 552 R0-intent surgeries for CRLM, Nishioka et al. found that width of resection margin does not influence local recurrence rate [21]. As HGP is a post-resection parameter, pre-operative surrogates for the HGP would need to be established for this to be taken into account in clinical treatment planning.

In the last 10 years, multiple studies have looked at assessing HGP as a prognostic biomarker. These studies have included both untreated patients with resectable metastatic disease and patients with preoperative chemotherapy. In both chemo-naïve and in preoperatively treated patients, the desmoplastic pattern has consistently emerged as the HGP with the most favorable outcome. Furthermore, any amount of non-desmoplastic growth negatively impacts outcome. However, remarkable progress has been made in biomarker-driven therapies for unresectable metastatic colorectal cancer, supporting re-evaluation of these patients. In addition to chemotherapy regimens used in CRLM, biological agents such as anti-VEGF, anti EGFR-antibodies, multikinase inhibitors, and immunotherapy with checkpoint inhibitors in patients with microsatellite instability (MSI-H) are now treatment options for primarily non-resectable CRLM. Only a few studies have investigated the role of HGP in the era of new biological agents, leaving prospects for future clinical validation studies on patients treated with novel preoperative strategies. 

## 7. Discussion

Currently, assessing histopathological growth patterns in CRLM provides information from a single time point in a selected group of patients with resectable/potentially resectable disease. In order to test the plasticity of growth patterns and identify optimal future treatments, a non-invasive, preoperative method that would allow determination of growth patterns at different timeframes is required. This would enable selection of patients with exclusively desmoplastic metastases for curative surgical resection alone, thus avoiding the undesirable side-effects of adjuvant chemotherapy. Moreover, patients with angiogenic desmoplastic growth pattern could be candidates for anti-angiogenic therapy in a neoadjuvant setting. 

Further insights in interactions between cancer cells and different components of the tumor microenvironment might be elucidated by single cell-RNA sequencing. This could provide a better understanding of particular cellular subsets with specific functions and spatial localization. Given the variation in the tumor immune microenvironment of each HGP, the selection of immunomodulatory therapy might need to be adapted to the growth pattern, resulting in treatment strategies for inoperable patients.

As the field of digital pathology is expanding and many pathology services are becoming increasingly digitized, implementing machine learning and artificial intelligence-based algorithms may become an aid in assessing HGPs and even finding novel relevant clinicopathological features [77]. Until now, most AI research performed on liver metastasis has been conducted on radiological images [78,79]; however, with the digitalization of histological images, more studies are expected in the future [77].

The field of spatial transcriptomics will allow further exploration of the mechanisms, cell-to-cell interactions and temporal alterations noted from primary to metastases, the involvement and contribution of the tumor microbiome, and the effects of chemotherapy on a complexity scale hitherto not possible [80,81,82]. Single-cell-level analyses of cellular mechanisms and interactions are already being explored in primary tumors and colorectal liver metastases [63,81,82]. 

In order to better understand tumor biology and test treatment options specific to each HGP, patient-derived xenografts and organoids were obtained. Although with the lack of adaptive immune response, these will not deliver complete information about the immune microenvironment associated with the different growth patterns. However, patient-derived explants maintain both histological architecture and immune response generated in human tumors. Several novel avenues for research are to be explored (Figure 3), including the roles of macrophages and technological improvements in image analyses and deep learning strategies [31].

## 8. Conclusions

Extended evidence supports integrating HGP in the pathological assessment of CRLM. Adhering to current guidelines for scoring growth patterns is key, as standardized methodology and nomenclature enables studies on large patient cohorts on the biological mechanisms of HGP. Eventually, increased insight into the mechanisms of cancer biology may yield new avenues for targeted and tailored patient treatment in CRLM.

## Figures and Tables

**Figure 1 cancers-15-00181-f001:**
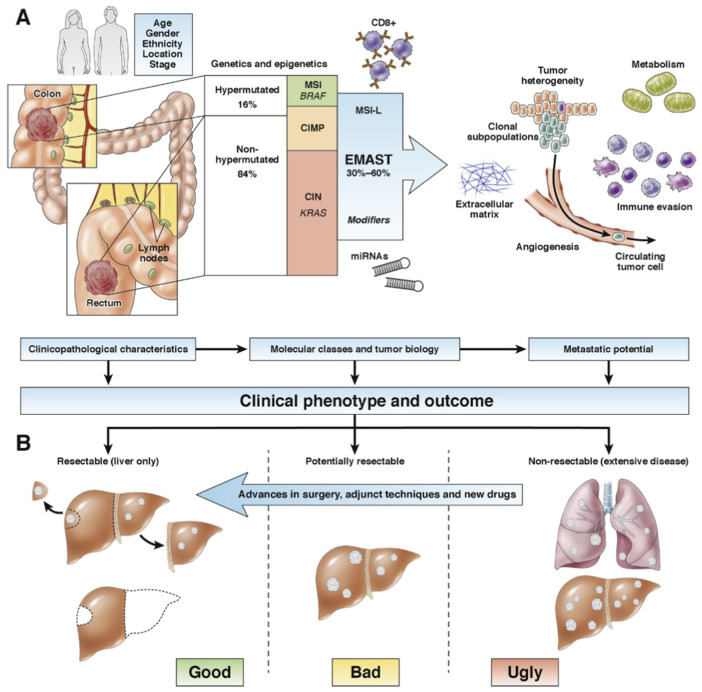
Clinical and molecular influence on the aggressiveness of colorectal liver metastasis. (**A**) Clinical behavior of colorectal cancer is determined by several factors, including demographic data (age, gender, and race), tumor presentation (location and stage), and timing of presentation of metastasis (synchronous or metachronous). Embedded in the cancer cells are the molecular pathways, which follow distinct forms of genomic instability yet with partly overlapping areas. Hypermutated cancers belong to the microsatellite instable (MSI) cancers and in part the CpG-island methylator phenotype (CIMP) cancers. Nonhypermutated cancers follow in large part the chromosomal instability (CIN)–driven pathways, often involving KRAS mutations from an early stage. The propensity to develop metastasis may possibly be modified through the elevated microsatellite alterations at selected tetranucleotide repeat (EMAST) and associated mechanisms, such as regulation of microRNAs or activity and numbers of CD8^+^ immune cells. Finally, the microenvironment contains numerous factors that may facilitate or propagate metastasis to invade, spread, and settle in new organ sites, particularly the liver and the lungs. (**B**) Determined by the clinical presentation, the genetic traits, and molecular mechanisms, the prognosis in colorectal liver metastasis is related to resectabilty for long-term survival. “Good” cases amenable for surgery have fewer bad genetic traits, such as less likelihood for BRAF mutations or KRAS mutations and were more likely to have EMAST and MSI-L alone in the current study. Patients with concomitant liver and lung metastases have an “ugly” tumor biology and are more likely to have higher frequencies of both KRAS and BRAF mutations and respond poorly to any line of treatment. The “bad” cases are in between, and the shift from “nonresectable” to “resectable” experiences a positive drift with time and where changing practice in surgical strategy, novel techniques, and use of conversion chemotherapy regimens improve outcomes. Novel biomarkers may aid in understanding aggressiveness of liver metastasis, assist in clinical decision making, and help to find new and more efficient therapies. Reproduced with permission from Soreide et al. Gastroenterology 2016 Apr;150(4):811-4, 2016 © AGA, published by Elsevier.

**Figure 2 cancers-15-00181-f002:**
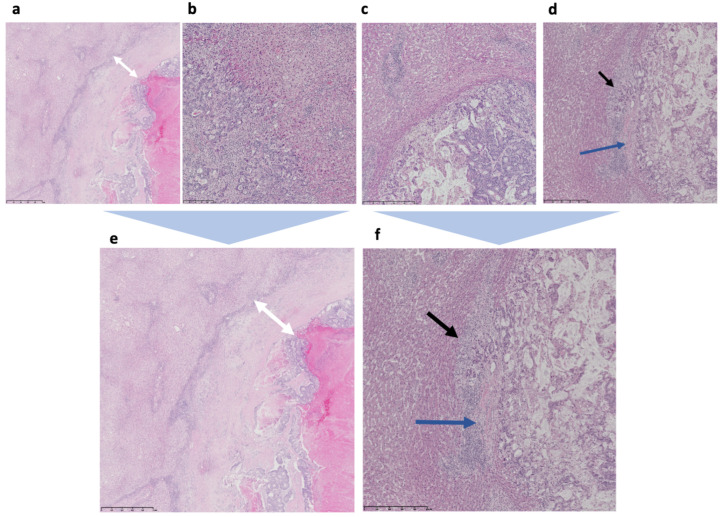
Histopathological growth patterns in a four-tier and two-tier system (H&E images, original data). (**a**) Low magnification of desmoplastic HGP, with white arrows marking the thick fibrous rim that separates the tumor from hepatocytes. (**b**) Low magnification of tumor–liver interface in replacement HGP with cancer cells in continuity with normal hepatocytes. Cancer cells form solid nests and trabeculae and there is no glandular differentiation. (**c**) Low magnification of pushing HGP with sharp interface between tumor cells and adjacent hepatocytes, without desmoplastic rim or tumor cells invading into liver tissue. (**d**) Low magnification of mixed/heterogeneous growth with replacement growth pattern (black arrow) and desmoplastic rim (blue arrow). (**e**) higher resolution of desmoplastic HGP, with white arrows marking the thick fibrous rim that separates the tumor from hepatocytes. (**f**) higher resolution of of mixed/heterogeneous growth with replacement growth pattern (black arrow) and desmoplastic rim (blue arrow). Images are provided by the authors’ clinical series.

**Figure 3 cancers-15-00181-f003:**
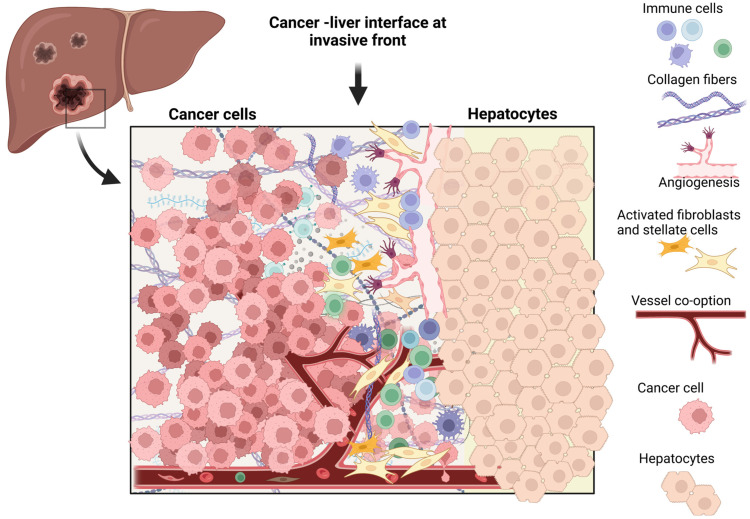
The tumor immune microenvironment in colorectal liver metastasis. The interface between tumor cells and the hepatocytes in colorectal liver metastasis, illustrating components of the tumor-immune microenvironment and other mechanisms that have been studied in the context of different histopathological growth patterns, including angiogenesis, vessel co-option, immune cells, and activated stromal elements. For a detailed description, please refer to the main body of the text. Created with Biorender.com.

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
