# Peer review of "Histopathological Growth Pattern in Colorectal Liver Metastasis and The Tumor Immune Microenvironment"

_cancers, 2022, doi:10.3390/cancers15010181_

Round 1

Reviewer 1 Report

In the present study, the authors provide a comprehensive review of the histopathologic growth pattern and tumor-immune-microenvironment in the context of colorectal liver metastases. The authors are to be commended for their thorough and detailed review of this topic. I also thought the supporting figures (in particular Figure 1 and 2) served to significantly enhance the written content. Excellent work. 

My biggest suggestions center around organization and presentation of the content, listed below. 

  • Some of the sections (in particular 3.3) could benefit from improved organization of content. For example, instead of listing results from various studies, it would be more helpful to readers to have organized, summarizing statements with supporting evidence from studies (e.g. statement #1 (evidence A, B, C); statement #2 (evidence D, E, F; etc).  
    • For example, for section 3.3, I would recommend either organizing the section by immune phenotype or by HGP type (or even by topic (i.e. vessel co-option, angiogenesis, Claudin/barriers, etc.)). The current section as is has a mix of all three and it is difficult to follow.  
  • For section 3.3, I would also recommend decreasing the amount of “basic science verbiage” - it is a little too dense for a review paper of this scope and is distracting 
  • For section 3.3, it would be helpful to briefly describe what each of the 3 phenotypes mean for people who are not familiar with them
  • Globally, the manuscript needs english grammar/syntax revisions

Author Response

Response to referee #1

#1

Comments and Suggestions for Authors

In the present study, the authors provide a comprehensive review of the histopathologic growth pattern and tumor-immune-microenvironment in the context of colorectal liver metastases. The authors are to be commended for their thorough and detailed review of this topic. I also thought the supporting figures (in particular Figure 1 and 2) served to significantly enhance the written content. Excellent work. 

RE: Thank you for the very favorable and kind comments. Much appreciated.

My biggest suggestions center around organization and presentation of the content, listed below. 

Some of the sections (in particular 3.3) could benefit from improved organization of content. For example, instead of listing results from various studies, it would be more helpful to readers to have organized, summarizing statements with supporting evidence from studies (e.g. statement #1 (evidence A, B, C); statement #2 (evidence D, E, F; etc).  

For example, for section 3.3, I would recommend either organizing the section by immune phenotype or by HGP type (or even by topic (i.e. vessel co-option, angiogenesis, Claudin/barriers, etc.)). The current section as is has a mix of all three and it is difficult to follow.

RE: Thank you for this suggestion. This section has been revised and structured according to both topic and HGP, also the content has been reorganized as advised.   

For section 3.3, I would also recommend decreasing the amount of “basic science verbiage” - it is a little too dense for a review paper of this scope and is distracting 

For section 3.3, it would be helpful to briefly describe what each of the 3 phenotypes mean for people who are not familiar with them

RE: Thank you for these comments. We have described briefly what each of the cancer immune phenotypes mean and tried to limit the amount of basic science vocabulary.

Globally, the manuscript needs english grammar/syntax revisions

RE: Linguistic revisions have been done throughout the manuscript.

Reviewer 2 Report

This review article nicely updated references in the literature to advance current knowledge about histopathological growth patterns and tumor  immune microenvironment in colorectal liver metastasises where the cancer biology, molecular mechanisms and clinical outcomes were detailed and supported despite the fact that there are still unknown issues need to be unveiled. They finally made a conclusion that, given the fact that histopathological growth pattern and the tumor immune microenvironment represent distinct subtypes of cancer biology related to clinical outcome and survival in each patient, the future focus should be on the cancer mechanisms and their translational aspects to novel treatment targets for precise therapy. Although this manuscript has been well-written, it would be better if the authors can briefly summarize and integrate the subtitles 3.2-3.5 into a table with appropriate references so that readers could better appreciate what messages and why the authors tried to convey.

Author Response

Response to referee 2

#2

Comments and Suggestions for Authors

This review article nicely updated references in the literature to advance current knowledge about histopathological growth patterns and tumor immune microenvironment in colorectal liver metastasises where the cancer biology, molecular mechanisms and clinical outcomes were detailed and supported despite the fact that there are still unknown issues need to be unveiled.

RE: thanks for the kind comments.

They finally made a conclusion that, given the fact that histopathological growth pattern and the tumor immune microenvironment represent distinct subtypes of cancer biology related to clinical outcome and survival in each patient, the future focus should be on the cancer mechanisms and their translational aspects to novel treatment targets for precise therapy.

Although this manuscript has been well-written, it would be better if the authors can briefly summarize and integrate the subtitles 3.2-3.5 into a table with appropriate references so that readers could better appreciate what messages and why the authors tried to convey.

RE: Thanks for the comment. We tried to facilitate this, but this was felt awkward and redundant to the text. Hence, we have made efforts to revise the text where needed, and hope the revised version better brings forward the message to the reader.

Should a table be felt to be necessary, we would be happy to try to facilitate this.

Reviewer 3 Report

Histopathological growth pattern in colorectal liver metastasis and the tumor-immune-microenvironment

It is a review on the relation between HGP, tumor-immune- microenvironment (TIME) and their relation with outcomes in CRLM. It is crisp, informative, and has good conclusions. Authors described various studies that looked at the HGP/TIME and the outcomes. They recognized the deficiencies in the current knowledge and brought up the use of AI which might be the future.

Please consider following to make the paper a better on

Simple Summary

-           Consider simplifying the language so non-medical personnel can understand it. Specifically, define histological growth pattern

Abstract

-          You are not expected to use the side headings – background, methods, results, and conclusions. Consider deleting it  

-          The references are not according to the journal. They should be at the end of the sentence and not in the middle. There should be space between the last letter of the sentence and the reference.

Introduction

-          Reference for the first line

Method

-          If this is a systematic review, consider adding the PRISMA scheme. If this is not a systemic review, remove this section.  

Results

-          Need reference for line 120

-          Line 122-127, content in these lines is repeated in the following section (in detail). Rewrite it.

-          Rewrite lines 136-138; the intent is not clear.

-          Line 141 – you need not expand HGP as it was done earlier.

-          Move figure 2 up, after line 161

-          Lines 176-179, can be added to the previous para.

-          Line 266, should it be HGP?

-          Line 337, did you mean to say, “escape”

Tables and figures

-          Add a table for section 3.2 for three HGP patterns. To the same table, you can add the relevant content from section 3.3

-          There are many studies the looked at outcomes based on HGP patterns – consider adding tables for it – it will be easy for the readers.

-          Figure 3, the order of the figure is not covering the entire figure. 

Author Response

Response to referee 3

#3

Comments and Suggestions for Authors

Histopathological growth pattern in colorectal liver metastasis and the tumor-immune-microenvironment

It is a review on the relation between HGP, tumor-immune- microenvironment (TIME) and their relation with outcomes in CRLM. It is crisp, informative, and has good conclusions. Authors described various studies that looked at the HGP/TIME and the outcomes. They recognized the deficiencies in the current knowledge and brought up the use of AI which might be the future.

Please consider following to make the paper a better on

Simple Summary

-           Consider simplifying the language so non-medical personnel can understand it. Specifically, define histological growth pattern

RE: Thank you for this suggestion. The histopathological growth pattern has been defined in the simple summary.

Abstract

-          You are not expected to use the side headings – background, methods, results, and conclusions. Consider deleting it  

RE: Thank you for this suggestion. We have deleted the side headings.

-          The references are not according to the journal. They should be at the end of the sentence and not in the middle. There should be space between the last letter of the sentence and the reference.

RE: Thank you for these comments. The spacing in the references has been revised accordingly.

Introduction

-          Reference for the first line

RE: Thank you for the comment. It has been added (1).

Method

-          If this is a systematic review, consider adding the PRISMA scheme. If this is not a systemic review, remove this section.

RE: Thank you for this suggestion. We have conducted a narrative/ literature review, not a systematic review, therefore the Methods section has been removed.   

Results

-          Need reference for line 120

RE: Thank you for the comment. References (24-27) have been provided.

-          Line 122-127, content in these lines is repeated in the following section (in detail). Rewrite it.

RE: Thank you for the comment. Some of the repeated content has been erased and the section has been rephrased.

-          Rewrite lines 136-138; the intent is not clear.

RE: Thank you for the comment. The above sections have been rephrased.

-          Line 141 – you need not expand HGP as it was done earlier.

-          Move figure 2 up, after line 161

RE: Thank you for the comments. As figure 2 is illustrative for content further that line 161, we chose to keep it in place to better illustrate the work progress between the 2 consensus guidelines articles. We believe this will be corrected for the final typesetting and PDF of the article anyway.

-          Lines 176-179, can be added to the previous para.

-          Line 266, should it be HGP?

-          Line 337, did you mean to say, “escape”

RE: Thank you for the comments. The suggested changes and corrections have been made.

Tables and figures

-          Add a table for section 3.2 for three HGP patterns. To the same table, you can add the relevant content from section 3.3

-          There are many studies the looked at outcomes based on HGP patterns – consider adding tables for it – it will be easy for the readers.

-          Figure 3, the order of the figure is not covering the entire figure. 

RE: Thanks for the suggestions. We believe the revisions have made the content more clear and adding further tabularized material would clutter the content. The consensus is now 2 main types of HGP.

Reviewer 4 Report

This is a well written review of the impact of histological growth pattern of colorectal liver metastases on clinical outcomes and its reasons behind including the genetic and molecular mechanisms. I have a few comments as below.

1.  In addition to desmoplastic reaction, the microenvironment indices such as poorly differentiated clusters or tumor budding have been reported as prognostic predictors. Could the authors comment on these?

2.  Please briefly describe the methodology of evaluating HGP especially which cut surface or slice of the tumor is evaluated.

3. Are there any knowledges on the difference in HGP between the micro-environment of primary colorectal cancer and liver metastases? I just wondered because it is quite common that metastatic tumors are less differentiated than the primary one.

4. Pleas present previously reported figures regarding prognosis of the patients with each category of HGP such as survival rate .

Author Response

Response to referee #4

Comments and Suggestions for Authors

This is a well written review of the impact of histological growth pattern of colorectal liver metastases on clinical outcomes and its reasons behind including the genetic and molecular mechanisms. I have a few comments as below.

  1. In addition to desmoplastic reaction, the microenvironment indices such as poorly differentiated clusters or tumor budding have been reported as prognostic predictors. Could the authors comment on these?

RE: Thank you for this suggestion. This has already briefly been commented on in the section entitled Need for better understanding the cancer biology of HGP in CRLM. Low tumor budding scores and Crohn’s disease-like response have been associated with desmoplastic HGP and better overall survival.

  1.  Please briefly describe the methodology of evaluating HGP especially which cut surface or slice of the tumor is evaluated.

RE: Thank you for this suggestion. A concise explanation of the methodology described in the first consensus guidelines is added to section 3.

  1. Are there any knowledges on the difference in HGP between the micro-environment of primary colorectal cancer and liver metastases? I just wondered because it is quite common that metastatic tumors are less differentiated than the primary one.

RE: Thank you for this question. Findings of recent studies are mentioned in section 4. However, the HGP is specific to the liver metastasis, and direct comparison to primary tumor is somewhat arbitrary.

  1. Pleas present previously reported figures regarding prognosis of the patients with each category of HGP such as survival rate .

RE: Thank you for this suggestion, some of the previously reported survival figures are listed in the text. A direct depiction of survival curves from previous publications would be somewhat arbitrary, and hence the figures are stated as text for readability.

Reviewer 5 Report

The Authors have performed an interesting and pleasant to read narrative review. Liver metastases from colorectal cancer are a highly relevant issue from both clinical and biological point of views. The work is well organized and presented in all parts. The narration is structured in order to inform the readers about correlation between prognosis of CRLMs and histopathological and the tumor-immune-microenvironment characteristics. I fail to find any minor/major flaws in this manuscript.

I have just some minor recommendations:

1. Please I suggest to add information about oligo-metastatic CRLMs. This can improve the clinical strength as well as the scientific completeness of the manuscript. An interesting basis to discuss and cite is PMID: 33096795

2. Please clarify if the histopathological sections showed in Figure 2 are original data or not.

Author Response

Response to referee 5

#5 Comments and Suggestions for Authors

The Authors have performed an interesting and pleasant to read narrative review. Liver metastases from colorectal cancer are a highly relevant issue from both clinical and biological point of views. The work is well organized and presented in all parts. The narration is structured in order to inform the readers about correlation between prognosis of CRLMs and histopathological and the tumor-immune-microenvironment characteristics. I fail to find any minor/major flaws in this manuscript.

RE: Thank you for your kind comments.

I have just some minor recommendations:

  1. Please I suggest to add information about oligo-metastatic CRLMs. This can improve the clinical strength as well as the scientific completeness of the manuscript. An interesting basis to discuss and cite is PMID: 33096795

RE: Thank you for this suggestion. The reference has been added (ref 68).

  1. Please clarify if the histopathological sections showed in Figure 2 are original data or not.

RE: Thank you for this comment. The images of histopathological sections in Figure 2 are original data from the authors’ own clinical series. We have now specified this in the legend.

Round 2

Reviewer 3 Report

Author's changes to the manuscript, but now they moved it to two classifications from 3. I still think a table summarizing the differences would go a long way but the text is easy to follow now.